# Towards Well-distributed Generative Networks Using Adversarial Autoencoders

## Abstract

In addition to perceptual quality, the usefulness of a generative model depends on how closely the generated distribution matches the training distribution. Previous efforts in adversarial generative models have focused on reducing "mode collapse", but this term, roughly meaning being unable to generate certain parts of the data distribution, is not clearly defined. In addition, being able to generate every image in the data distribution does not imply reproducing the correct distribution, which additionally requires that each image occur at the same frequency in the generated images as in the training data. Due to the lack of a precise definition and measurement, it has been difficult to evaluate the success of these efforts in producing the correct distribution. In this work we proposes an autoencoder-based adversarial training framework, which ensures that the density of the encoder's aggregate output distribution closely matches the prior latent distribution, which in turn ensures that the distribution of images generated from randomly sampled latent code will closely match the training data. To evaluate our method, we introduce the 3DShapeHD dataset, which has a moderate complexity that goes beyond simplistic toy datasets, but also a exactly known generating process and distribution of features, which enables precise measurements. Using the reduced chi-square statistic, we show significant improvement in the accuracy of the distribution of generated samples. The results also demonstrate that the enhanced diversity of our model improves the ability to generate uncommon features in real-world datasets.

## 1 Introduction

For generative models, perceptual quality of the output is usually of high interest. However, being able to generate nice-looking images does not guarantee that the data distribution is accurately modeled. It has been previously observed and discussed (Bau et al., 2019; Arora & Zhang, 2017; Arora et al., 2017) that the objective of GANs is "unable to prevent mode collapse and mode dropping", and fails to ensure the generated data is in the target distribution. Nonetheless, matching the training and generated distribution of the data can be crucial. Generative models have been used for data augmentation, and failure to reproduce the original data distribution faithfully could lead to undesired bias. Also, failing to learn diversity in the training distribution will lead to the failure in the inversion task, which is often used for manipulations (Bau et al., 2020; Brock et al., 2016; Zhu et al., 2016), image editing (Shen et al., 2020), and unsupervised learning (Donahue et al., 2016; Dumoulin et al., 2016).

The foundational theories on GANs (Goodfellow et al., 2014) were developed under the assumptions that the networks had unlimited capacity. However, doubts were raised by the authors in Arora et al. (2017), who provided theoretical analysis on whether the same applies when the discriminator has a bounded size. Arora & Zhang (2017) and Bau et al. (2019) further design testing benchmarks and metrics on the support size (which roughly means the number of visually distinct images that can be generated) in the distribution and provide evidence that the size of the support of the generated distribution scales near-linearly with discriminator capacity.

A large amount of efforts have been made to reduce mode collapse, for example, by regularizing the discriminators (Metz et al., 2016; Arjovsky et al., 2017; Mao et al., 2017; Salimans et al., 2016; Lin

et al., 2018) or introducing auxiliary networks to encoders (Arjovsky & Bottou, 2017; Srivastava et al., 2017; Mao et al., 2019).

While all of the above approaches show some improvement, fundamental difficulties remain. Indeed, we feel that the term "mode collapse" itself does not have a widely received meaning, which in turn means that it is not possible to devise a proper measurement to precisely quantify the extent of the supposed mode collapse. Existing metrics on general image datasets typically use deep feature statistics, e.g. Frechet Inception Distance (FID) and inception score, which mix image quality and diversity in a single hard-to-interpret number.

Whenever more direct assessments do exist, they are generally limited to very simple toy datasets. For example, in Unrolled GAN (Metz et al., 2016) the density of the generated distribution is directly plotted, but the dataset is just a set of two-dimensional points drawn from a mixture of Gaussians. It is questionable whether such assessment has practical relevance, as in typical use cases like images, the data representation has hundreds of thousands to millions of dimensions.

The other problem with the notion of "mode collapse" is that it relates to the support of the generated distribution, but not the density. If a generator is able to generate every image in the data distribution, it might be regarded as free from mode collapse, but this does not mean that each possible image occur with the same frequency in randomly generated samples as in the training data.

On the other hand, autoencoder-based methods like the Variational Autoencoder (Kingma & Welling, 2013), which naturally do not suffer from mode collapse since they are trained to generate every image in the training set through reconstruction, receive relatively little attention, since their inferior perceptual quality severely limits their utility. Nevertheless, we do find the collapse-free nature of autoencoders appealing as a foundation to build upon. We propose a novel adversarial training strategy that ensures a close match between the aggregate posterior distribution of the encoder and the prior latent distribution. This result is that random samples generated from the prior latent distribution will be distributed correctly.

The problem then reduces to replicating the same structure of the latent space in a high-quality generator, which we solve through our proposed autoencoder-coupled GAN.

To address the difficulty in evaluation, we propose using the 3DShapesHD dataset, which is our enhanced version of the 3DShapes dataset (Burgess & Kim, 2018) originally designed for disentanglement learning, for a precise measurement of the sample distribution of generated models under a more realistic setting. It is a synthetic image dataset with a precisely known generating process and distribution of generating parameters. While having a complexity transcending simple toy datasets, it is nevertheless simple enough to enable mathematically well-founded quantitative evaluation using goodness-of-fit tests.

To summarize, this work makes the following contributions:

- A training framework for autoencoders that the aggregate posterior distribution matches the prior distribution.
- A method for coupling an trained autoencoder with a GAN with only minimal modification to the optimization procedure of GANs, which transfers the well-structured latent space of the autoencoder to the GAN without introducing the blurriness, thus reinforcing the high-quality perceptual quality of GANs with the correct distribution of autoencoders.
- A procedure for testing the correctness of sample distribution of generative models using goodness-of-fit test on a dataset with known distribution, along with a suitable dataset.

## 2 RELATED WORKS

**Generative Models for Image Generation** The complexity of images presents unique difficulties for probabilistic modeling techniques. Generative Adversarial Networks (GAN) (Goodfellow et al., 2020) enable efficient sampling of high-resolution images with good perceptual quality (Brock et al., 2018; Karras et al., 2020), but are difficult to optimize (Gulrajani et al., 2017; Mescheder, 2018; Arjovsky et al., 2017; Miyato et al., 2018) and struggle to capture the full data distribution (Metz et al., 2016). In contrast, likelihood-based methods focus on good density estimation, which makes optimization more well-behaved. Variational autoencoders (VAE) (Kingma & Welling, 2013) and flow-

based models (Dinh et al., 2014; 2016) allow efficient synthesis of high-resolution images (Child, 2020; Kingma & Dhariwal, 2018; Vahdat & Kautz, 2020), but sample quality does not match GANs. While autoregressive models (ARM) (Van Den Oord & Dieleman, 2016; Van den Oord et al., 2016; Child et al., 2019; Chen et al., 2020) achieve strong performance in density estimation, computationally demanding architectures (Vaswani et al., 2017) and a sequential sampling process limit them to low-resolution images. Since pixel-based representations of images contain barely perceptible, high-frequency details (Salimans et al., 2017), maximum-likelihood training spends a disproportionate amount of capacity modeling them, resulting in long training times. Diffusion probabilistic models (Sohl-Dickstein et al., 2015) have produced top results for density estimation (Kingma et al., 2021) and sample quality (Dhariwal & Nichol, 2021). These models are so good at generating images because their neural networks are built like U-Nets (Dhariwal & Nichol, 2021; Ho et al., 2020; Ronneberger et al., 2015; Song et al., 2020). Diffusion Models usually generate the highest-quality samples when trained using a reweighted objective function. Diffusion Models have recently been applied to high-resolution image generation with promising results. For example, Rombach et al. (2022) proposes a latent diffusion model for synthesizing photorealistic high-resolution images.

To improve the limitations of individual generative methods, much research (Dai & Wipf, 2019; Esser et al., 2021; Razavi et al., 2019; Rombach et al., 2020; Yan et al., 2021; Yu et al., 2021) has focused on combining the strengths of different techniques into more efficient and higher-performing models using a two-stage approach. VQ-VAEs (Razavi et al., 2019; Yan et al., 2021) use autoregressive models to learn an expressive prior over a discretized latent space. Ramesh et al. (2021) extends this approach to text-to-image generation by learning a joint distribution over the discretized image and text representations. More broadly, Rombach et al. (2020) uses conditionally invertible networks to provide a generic transfer between latent spaces of diverse domains. Besides, VQGANs (Esser et al., 2021; Yu et al., 2021) use a first stage with an adversarial and perceptual objective to scale autoregressive transformers to larger images.

Recently, StyleGAN-XL (Sauer et al., 2022) has achieved state-of-the-art results on many image generation tasks. The synthesis network of StyleGAN-XL achieves translation equivariance by using the alias-free primitive operations of StyleGAN3 (Karras et al., 2021). It is trained progressively, increasing the output resolution over time by introducing new synthesis layers. The discriminator structure remains unchanged during training, and the early low-resolution images are upsampled as needed. Additionally, the already trained synthesis layers are frozen as new layers are added.

**Mode Collapse** GANs utilize a generator and a discriminator engaged in an adversarial game. At convergence, the generator learns to generate photorealistic images. Despite their remarkable success, GANs suffer from the major problem of mode collapse (Arjovsky & Bottou, 2017; Zhang et al., 2017; Chen et al., 2016; Metz et al., 2016; Salimans et al., 2016). While theoretically, convergence guarantees the generator learns the true data distribution, in practice reaching equilibrium is difficult and not guaranteed, potentially leading to mode collapse (Arora & Zhang, 2017; Santurkar et al., 2018; Bau et al., 2019).

Some methods focus on improving the discriminator with different optimization techniques (Metz et al., 2016; Zhang et al., 2021) and divergence metrics (Arjovsky et al., 2017; Mao et al., 2017) to stabilize training. The minibatch discrimination approach (Salimans et al., 2016) allows the discriminator to distinguish between whole mini-batches of samples instead of between individual samples. Durugkar et al. (2016) use multiple discriminators to address this. Other methods use auxiliary networks to reduce mode collapse. ModeGAN (Che et al., 2016) and VEE-GAN (Srivastava et al., 2017) enforce the one-to-one mapping between input noise vectors and generated images with extra encoder networks. Multiple generators (Ghosh et al., 2018; Hoang et al., 2018) and weight-sharing generators (Liu & Tuzel, 2016) capture more modes of distribution but entail heavy compute costs or modify the network design. PacGAN (Lin et al., 2018) modifies the discriminator to make decisions based on multiple samples from the same class. Mode-seeking GAN (Mao et al., 2019) uses a novel objective function that encourages the generator to generate diverse images by seeking out the modes of real data distribution. Zhang et al. (2019); Miyato et al. (2018) introduce a spectral normalization technique to stabilize the training of the networks and mitigate mode collapse. Dropout-GAN (Mordido et al., 2018) uses dropout regularization to encourage the generator to learn a more diverse set of features and reduces the impact of mode collapse.

## 3  DATASET AND EVALUATION METHOD

In the few existing works where direct measurements or visualizations of the generated distribution are available, the datasets generally consist of unstructured points in low-dimensional spaces, drawn from relatively simple distributions, in stark contrast with typical use cases where the data is high-dimensional, has a specific modality (e.g. images) and a complex distribution. We wish to evaluate our method on a dataset with a suitable modality and non-trivial structure. We choose the 3DShapes dataset for this purpose.

### 3.1  CONSTRUCTING THE DATASET

The original 3DShapes dataset (Burgess & Kim, 2018) consists of $64 \times 64$ images, each depicting a variation of a simple scene. The scene consists of a square box containing a geometric object which can be a cube, a cylinder, a sphere, or a capsule (a cylinder plus two hemispherical caps). The camera is at the same height as, and points at, the object. The object, floor, and wall are each colored using one of 10 different hues. The object comes in 8 different sizes, and the camera can be placed at 15 different angles. The dataset includes one image for every possible combination of these variables, giving a total of $4 \times 8 \times 10^3 \times 15 = 480,000$ images.

This dataset is moderately complex, having 6 generating factors that interact nontrivially. It also features a mixture of qualitatively different factors: linear (size, view angle), discrete (shape), and continuous with a nontrivial topology (the hues, with the topology of a circle). It is thus closer to real-world scenarios than simpler toy datasets, yet its actual generating process and distribution of factors are precisely known, allowing for direct measurements.

However, the original dataset has several drawbacks: first, the resolution is far below the capacity of modern generative models; second, the continuous factors are only sampled at a small set of discrete values; and last, the dataset is a perfectly stratified set of samples, while in-the-wild datasets are generally not stratified. So we created a high-resolution ($256 \times 256$) version of this dataset, with the generating factors of each image sampled i.i.d. uniformly from the entire range. Our dataset consists of $500,000$ images made using the POV-Ray renderer (pov, 2021). Examples from our dataset are shown in figure 3.

### 3.2  MEASURING THE OUTPUT DISTRIBUTION

Since the generating factors in the dataset are uniformly distributed, an ideal generator trained on this dataset should also generate images with uniformly distributed factors. We train a recognition network that computes the six generating factors from an image. Then, we divide the range of values of the generating factors into equal-sized bins, take the generator, generate a large number of random samples, and place them into the bins using the recognition network. We then perform a goodness-of-fit test to see whether the uniform distribution is a good fit to the distribution generated by the generator, by computing the reduced chi-square statistic of the number of samples in each bin. If there are $k$ bins and $n$ samples are generated, each bin is expected to contain $\frac{n}{k}$ samples. The degree of freedom is $\nu = k - 1$, and the reduced chi-square statistic, or chi-square per degree of freedom, is

$$\chi_\nu^2 = \frac{1}{k-1} \sum_{i=1}^{k} \frac{(n_i - \frac{n}{k})^2}{\frac{n}{k}} \tag{1}$$

where $n_i$ is the number of samples in bin $i$. A smaller value indicates that the uniform distribution is a good fit for the observed sample distribution. Since the theoretical distribution has no parameter, there is no possibility of over-fitting, and $\chi_\nu^2$ should be no smaller than 1.

We divide the range of values of each factor into bins as follows: the three hue angles are each divided into 12 bins of $30°$; the maximum object size is twice the minimum and the range of relative scale $[1, 2]$ is divided into four bins of length $0.25$; the view angle is in the range $[-30°, 30°]$, divided into 6 bins of $10°$; each of four shapes is a separate bin. Combining these, we divide the space of generating factors into $k = 4^2 \times 6 \times 12^3 = 165,888$ equal-sized bins. We generate $n = 20k$ samples.

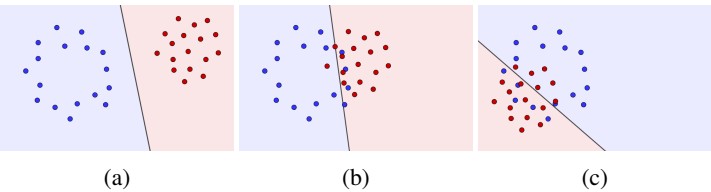

(a)            (b)            (c)

Figure 1: Illustration of oscillation

## 4 THE WELL-DISTRIBUTED GENERATOR

We start from autoencoders, which are inherently resistant against mode collapse. We aim to improve it further so that it is not just collapse-free, but actually produces a posterior latent distribution that closely matches the prior. This then implies that samples generated from the prior will match the data distribution, except that they will be highly blurred. Then, we use a coupling technique to replicate the latent distribution of the autoencoder in a GAN.

Our autoencoder part is based on the Adversarial Autoencoder (AAE) (Makhzani et al., 2015). The AAE consists of an encoder $E$, a generator $G$, and a code discriminator $C$. The $E$ and $G$ are trained to reconstruct the input image. At the same time, $C$ is trained to distinguish between the output of the encoder, $E(\mathbf{x})$, where $\mathbf{x} \sim p(\mathbf{x})$ and $p(\mathbf{x})$ is the data distribution, and random codes drawn from the prior latent distribution $p(\mathbf{z})$. In addition to reconstruction, $E$ is trained to make its output indistinguishable by $C$ from the prior distribution.

We choose the standard normal distribution as the prior. Makhzani et al. (2015) discusses three configurations: the deterministic encoder, the parametrized posterior where the encoder produces the parameter of a family of posterior distributions and sample from it, and the universal approximator posterior where the encoder has access to some noise. The original AAE uses the deterministic option. We combine the latter two options: in case the dimensionality of the data manifold is lower than the chosen prior, additional noise is required to boost the dimensionality of the encoder's input, since a deterministic encoder cannot transform a low-dimensional data distribution to match a high-dimensional prior. The universal approximator posterior achieves this with more flexibility, but for reasons that will be discussed later, a parametrized posterior is also desirable.

Our posterior and noise are also the normal distribution. We subsume the two additional sources of randomness (noise and sampling) into a single noise vector $\boldsymbol{\epsilon}$. Thus, the objective of our formulation of a basic AAE is

$$\min_{C} \left( \mathop{\mathbb{E}}_{\boldsymbol{z} \sim \mathcal{N}(\boldsymbol{0},I)} \big[ -\ln C(\boldsymbol{z}) \big] + \mathop{\mathbb{E}}_{\substack{\boldsymbol{x} \sim p(\boldsymbol{x}) \\ \boldsymbol{\epsilon} \sim \mathcal{N}(\boldsymbol{0},I)}} \big[ -\ln(1 - C(E(\boldsymbol{x}, \boldsymbol{\epsilon}))) \big] \right) \tag{2}$$

$$\min_{E,G} \mathop{\mathbb{E}}_{\substack{\boldsymbol{x} \sim p(\boldsymbol{x}) \\ \boldsymbol{\epsilon} \sim \mathcal{N}(\boldsymbol{0},I)}} \big[ \|G(E(\boldsymbol{x}, \boldsymbol{\epsilon})) - \boldsymbol{x}\| - \lambda_C \ln C(E(\boldsymbol{x}, \boldsymbol{\epsilon})) \big] \tag{3}$$

where $\lambda_C$ is a weighting hyperparameter. Theoretically, this simple procedure should suffice to ensure that the aggregate posterior distribution of the latent code is identical to the prior. However, in practice it proved inadequate, in that it is hampered by strong oscillation.

Oscillation has been a chronic problem in adversarial networks. The visual manifestation is that the generated samples will display a general trend, which changes rapidly during training. For example, consider training on a set of portrait photos and tracking the change of generated images during training: after a short period, everyone starts smiling; then after another short period, everyone turns their head to the left; then after another short period, everyone gains more hair, etc.

Figure 1 shows a greatly simplified illustration of this problem: the blue points are the training samples, and the red ones are generated. The black line is the decision boundary of the discriminator (a). The generator pushes the generated samples towards the training samples (b), but will often narrowly miss the correct distribution either because it is overshot or because the gradient given by the discriminator is not optimal (c). The decision boundary of the discriminator suddenly changes, usually resulting in it being non-optimal, and the process repeats.

We introduce two modifications to ensure a more stable training. Once the oscillation is alleviated, the result is in much better agreement with the theoretical effectiveness of this method.

## 4.1 SAMPLE DIFFERENCE DISCRIMINATOR

We exploit a property of the normal distribution. The Cramér's decomposition theorem (Cramér, 1936) says that, if the sum of two independent random variables x and y is normally distributed, then x and y are both normally distributed.

Now suppose that x and y are i.i.d. random variables, then x and $-$y are also independent, $\mathbb{E}[x-y] = \mathbb{E}[x] - \mathbb{E}[y] = 0$ and $\text{Var}[x - y] = \text{Var}[x] + \text{Var}[-y] = 2\text{Var}[x]$. If $x - y$ is normally distributed, then by the Cramér's decomposition theorem, x is also normally distributed, with half the variance of $x - y$.

That is, if x and y are i.i.d. random variables and $x - y$ is normally distributed, then given the distribution of $x - y$, the distribution of x is determined up to translation.

This can easily be generalized to random vectors: by definition, a random vector $\mathbf{x}$ is normally distributed if and only if every linear combination of its components is normally distributed. So, if $\mathbf{x} - \mathbf{y}$ is normally distributed, then every linear combination of its components is normally distributed. By the Cramér's decomposition theorem, these same linear combinations of the component of $\mathbf{x}$ are also normally distributed, which implies that $\mathbf{x}$ is normally distributed.

Note that $\mathbf{x} - \mathbf{y}$ will always have a mean of $0$ and thus cannot oscillate. So, if instead of $E(\boldsymbol{x}, \boldsymbol{\epsilon})$ the code discriminator receives $E(\boldsymbol{x}, \boldsymbol{\epsilon}) - E(\boldsymbol{x}', \boldsymbol{\epsilon}')$ as the input where $\boldsymbol{x}, \boldsymbol{x}', \boldsymbol{\epsilon}$ and $\boldsymbol{\epsilon}'$ are independent, then it ensures that $E(\boldsymbol{x}, \boldsymbol{\epsilon})$ is normally distributed while also preventing oscillation.

The remaining problem is that given that $\mathbf{x}$ and $\mathbf{y}$ are i.i.d. random vectors and $\mathbf{x} - \mathbf{y}$ is normally distributed, the distribution of $\mathbf{x}$ can be determined only up to translation, and there is no constraint on its mean. This can be solved in different ways, e.g. explicitly penalizing the absolute value of the sample mean of $E(\boldsymbol{x}, \boldsymbol{\epsilon})$ of a training batch. We elect to split the code discriminator into two, an MLP $C_1$ which receives the difference between two codes as the input, and a linear classifier $C_2$ consisting of just an affine function followed by sigmoid, which receives the code itself. The linear classifier is simple enough that it is stable and does not cause oscillation.

While this method is similar to batch normalization in that in both methods a batch of samples is transformed so that the sample mean becomes zero, we note several differences: under our choice of the normal distribution for our prior, the correctness of our method can be formally established, while for batch normalization in general, we are not aware of analogous results: if $x_1, \ldots, x_n$ are i.i.d. random variables that are not necessarily normally distributed, then given the distribution of $x_1 - \frac{1}{n} \sum_{i=1}^{n} x_i$, it is not known whether the distribution of $x_1$ can be determined up to translation and reflection in general. Our sample difference transformation is applied only before the network, while batch normalization is used throughout the network. The features in the hidden layers of the network are generally not normally distributed, so there is the risk that using batch normalization in the middle may cause the discriminator to fail to distinguish two distributions that are non-identical up to simple transformations.

## 4.2 SEPARATE TRAINING OF CONVOLUTION AND FULLY CONNECTED LAYERS

A larger batch size is generally beneficial in the highly unstable adversarial training procedure. While computing power is increasingly available, the capacity of generative models grow in tandem, and the use of very large batch sizes remains costly. By separating the training of the convolution part and the fully-connected part of the network, we can easily increase the batch size of the adversarial training to the thousands, thus greatly improving the stability of training.

We noted that, in terms of computational cost, the bottleneck is the convolution, and the fully-connected layers generally take up a tiny portion of computation time. If we are given the optimal weights of the convolution layers in $E$ in advance, then only the fully-connected part needs to be adversarially trained. Since the code discriminators do not contain convolution layers either, the adversarial part can be computed very efficiently, enabling the use of a much larger batch size.

By adding a few more layers in the fully-connected part of $E$ we can ensure that it is powerful enough to transform whatever feature distribution produced by the convolution layers into the desired prior distribution, yet still computationally cheap. Then, the convolution layers only need to be trained as a feature extractor.

We refer to all the convolution layers plus the first fully-connected layer of $E$ as the "trunk" and the rest as the "head". The noise is injected between the trunk and the head, so that the trunk is fully deterministic. We pre-train $E$ and $G$ as a VAE first, then freeze the trunk of $E$ and pre-compute and save the output features of the trunk on the whole dataset. Then, the discriminator loss and adversarial loss become fast to compute, and very large batches can be used. The reconstruction loss involves $G$ which contains convolution, so it can only be computed on small batches, but this is fine since reconstruction loss is stable.

The full training procedure of our well-distributed autoencoder thus consists of two phases:

**Phase 1: Variational Autoencoder.** In this phase, $E$ and $G$ are trained as a VAE. After training, the trunk of $E$ is frozen, and the trunk output feature is computed for the whole dataset and saved.

**Phase 2: Adversarial Autoencoder.** We optimize

$$\min_{C_1} \left( \mathbb{E}_{\boldsymbol{z} \sim \mathcal{N}(\boldsymbol{0}, 2I)} \left[ -\ln C_1(\boldsymbol{z}) \right] + \mathbb{E}_{\substack{\boldsymbol{x}, \boldsymbol{x}' \sim p(\boldsymbol{x}) \\ \boldsymbol{\epsilon}, \boldsymbol{\epsilon}' \sim \mathcal{N}(\boldsymbol{0}, I)}} \left[ -\ln(1 - C_1(E(\boldsymbol{x}, \boldsymbol{\epsilon}) - E(\boldsymbol{x}', \boldsymbol{\epsilon}'))) \right] \right) \tag{4}$$

$$\min_{C_2} \left( \mathbb{E}_{\boldsymbol{z} \sim \mathcal{N}(\boldsymbol{0}, I)} \left[ -\ln C_2(\boldsymbol{z}) \right] + \mathbb{E}_{\substack{\boldsymbol{x} \sim p(\boldsymbol{x}) \\ \boldsymbol{\epsilon} \sim \mathcal{N}(\boldsymbol{0}, I)}} \left[ -\ln(1 - C_2(E(\boldsymbol{x}, \boldsymbol{\epsilon}))) \right] \right) \tag{5}$$

$$\min_{E, G} \mathbb{E}_{\substack{\boldsymbol{x}, \boldsymbol{x}' \sim p(\boldsymbol{x}) \\ \boldsymbol{\epsilon}, \boldsymbol{\epsilon}' \sim \mathcal{N}(\boldsymbol{0}, I)}} \left[ \left\| G(E(\boldsymbol{x}, \boldsymbol{\epsilon})) - \boldsymbol{x} \right\| - \lambda_1 \ln C_1(E(\boldsymbol{x}, \boldsymbol{\epsilon}) - E(\boldsymbol{x}', \boldsymbol{\epsilon}')) - \lambda_2 \ln C_2(E(\boldsymbol{x}, \boldsymbol{\epsilon})) \right] \tag{6}$$

Where the L2-norm term is computed on small batches and everything else is computed on large batches. The weighting hyperparameters $\lambda_1$ and $\lambda_2$ are adjusted dynamically during training. Details are explained in the supplementary material.

### 4.3 Autoencoder-coupled GANs

So far, we have trained an autoencoder whose aggregate posterior distribution matches the prior distribution closely. But image autoencoders generally suffer from poor visual quality. To obtain a generator that could produce images that are both high-quality and properly distributed, our method will need to be combined with a high-quality generative model. In any case, we would like to retain the correspondence between images and latent codes learned by our autoencoder.

Let $G'$ be the GAN generator. The goal is for $G'(\boldsymbol{z})$ to be high-quality but also resemble $G(\boldsymbol{z})$, so that the well-structured latent space of $G$ can be transferred to $G'$. To avoid any possible degradation of image quality due to blurriness, we don't allow the use of an explicit reconstruction loss. Our solution is to use a conditional discriminator $D$ that takes pairs of images concatenated along the channel dimension as input. Each "positive" sample is a training image $\boldsymbol{x}$ paired with its reconstruction by the autoencoder $G(E(\boldsymbol{x}, \boldsymbol{\epsilon}))$, while each "negative" sample consists of $G'(\boldsymbol{z})$ and $G(\boldsymbol{z})$, the images generated by the GAN generator and the autoencoder generator from the same code $\boldsymbol{z}$. The two images in each positive pair resemble each other, so the discriminator enforces the same property on the negative pairs. The objective is

$$\min_{D} \left( \mathbb{E}_{\substack{\boldsymbol{x} \sim p(\boldsymbol{x}) \\ \boldsymbol{\epsilon} \sim \mathcal{N}(\boldsymbol{0}, I)}} \left[ -\ln D(\boldsymbol{x}, G(E(\boldsymbol{x}, \boldsymbol{\epsilon}))) \right] + \mathbb{E}_{\boldsymbol{z} \sim \mathcal{N}(\boldsymbol{0}, I)} \left[ -\ln(1 - D(G'(\boldsymbol{z}), G(\boldsymbol{z}))) \right] \right) \tag{7}$$

$$\min_{G'} \mathbb{E}_{\boldsymbol{z} \sim \mathcal{N}(\boldsymbol{0}, I)} \left[ -\ln D(G'(\boldsymbol{z}), G(\boldsymbol{z})) \right] \tag{8}$$

| Method | $\chi_\nu^2$ | FID | precision | recall |
|---|---|---|---|---|
| Ours+StyleGAN2 | 1.66 | 8.41 | 0.00 | 0.00 |
| AAE+StyleGAN2 | 2.81 | 6.06 | 0.00 | 0.00 |
| StyleGAN2 | 3.33 | 7.29 | 0.00 | 0.00 |
| StyleGAN-XL | 3.35 | 1.46 | 0.00 | 0.00 |
| (Theoretical) | 1.00 | 0.07 | 0.93 | 0.93 |

Table 1: Quantitative evaluation using reduced chi-square statistic, FID, precision and recall.

## 5 EXPERIMENTS

### 5.1 SETUP

For combining our method with GANs, if the autoencoder generator $G$ and the GAN generator $G'$ have the same architecture, then $G$ can be used as the initial weight of $G'$, which eases GAN training greatly. We take this option and use StyleGAN2 (Karras et al., 2020) for both the GAN generator and the autoencoder generator, even though the architecture is not tailored specifically to autoencoders. Naturally, we take the plain StyleGAN2 as a baseline.

We also adopt the gradient penalty, path penalty, and adaptive augmentation from StyleGAN2.

To validate our improvements on the AAE, we train a plain AAE according to equations 2 and 3 and combine it with GAN in the same way.

Additionally, we compare with StyleGAN-XL, which is representative of recent developments in the StyleGAN family of architectures.

For completeness, in addition to our reduced chi-square statistics, we also compute FID and precision/recall as defined in Kynkäänniemi et al. (2019). However, we found that these metrics correlate poorly with the reduced chi-square, which shows that they are not suitable for exact measurements.

Besides the 3DShapesHD dataset, we also trained our models on the FFHQ and LSUN Church datasets. Since the reduced chi-square statistic relies on knowing the true generating process and parameters which are not available, and the other metrics are unsuitable for quantitative evaluation, we only show some qualitative results. These are presented in the supplementary material.

### 5.2 QUANTITATIVE RESULTS

The reduced chi-square, FID, precision and recall of the trained models are compared in table 1. The FID, precision and recall are computed from $100,000$ generated samples and a subset of the dataset consisting of $100,000$ images. The "Theoretical" row gives the best possible performance: the FID, precision and recall are computed from two disjoint subsets of the dataset consisting of $100,000$ images each. Using our method, the excess reduced chi-square (that is, $\chi_\nu^2 - 1$, since 1 is the theoretical minimum) is reduced by more than $70\%$, compared to plain StyleGAN2.

We can see that the precision and recall of all models are zero, which means the behavior of these metrics on our dataset is pathological. The FID also correlates poorly with the reduced chi-square statistic. We discuss why they fail to give a meaningful evaluation of the models in the supplementary material.

To better visualize the result, in figure 2 we plot the number of bins containing exactly $k$ test samples for small $k$. The theoretical distribution is a binomial distribution. The other methods have much heavier tails than our method, relative to the theoretical distribution.

### 5.3 QUALITATIVE RESULTS

We probe the distribution of generated images of each model by projecting test images into the latent space of the generators. As can be seen in figure 2, the plain StyleGAN2 produces some empty bins, meaning that a small part of the data distribution cannot be generated. In other words, there is a

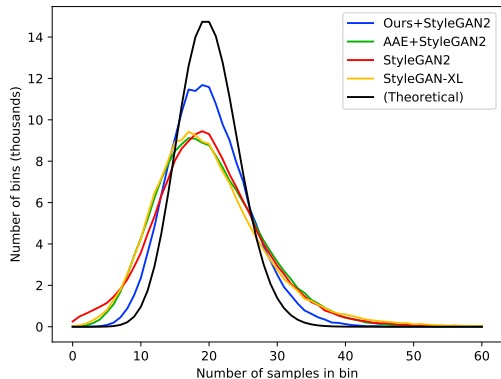

Figure 2: Distribution of bin frequency.

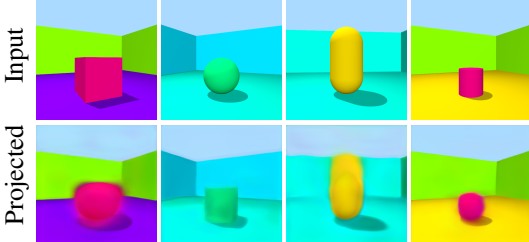

Figure 3: A few images that cannot be generated properly by the plain StyleGAN2 model.

slight mode collapse. Examples of such images are shown in figure 3. In contrast, our method does not produce empty bins.

We do note however, that modeling the density properly is a stronger property than just avoiding mode collapse: for all methods except plain StyleGAN2 there are no empty bins, which means each of them can generate every image in the distribution, yet our method approximates the data distribution more closely.

## 6 CONCLUSION

In this work, we propose a novel adversarial training procedure for autoencoders that ensures that the aggregate posterior latent distribution closely matches a standard normal distribution, thus ensuring that the distribution of samples generated from normally distributed latent codes closely matches that of the training data. By combining our autoencoder with GANs, we obtain generative models with accurate distribution as well as good visual quality. To evaluate our method, we introduce the 3DShapesHD dataset, a moderately complex dataset with exactly known generating procedure and distribution of parameters, so that the distribution of samples generated by a model trained on this dataset can be precisely compared with the true data distribution by a goodness-of-fit test using the reduced chi-square statistic. Our method greatly improves the accuracy of the distribution of randomly generated samples.

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
