# OpenReview forum: "Towards Well-distributed Generative Networks Using Adversarial Autoencoders"
_ICLR.cc/2024/Conference — ICLR 2024 Conference Withdrawn Submission_

### Official Review · Reviewer_EFy9 · 2023-10-23

**Soundness:** 2 fair
**Presentation:** 2 fair
**Contribution:** 1 poor
**Rating:** 3
**Confidence:** 4

**Summary:**

This paper first attempts to provide a more precise definition of the well-known mode collapse issue in adversarial generative models, with additional probabilistic guarantees. It further proposes an autoencoder-based adversarial training framework comprising an adversarial autoencoder like optimization process and a GAN-like optimization process. The framework is claimed to better ensure the statistical similarity of the generated distribution from randomly sampled latent codes to the training data. In addition, the 3DShapeHD dataset is proposed as an extension of the 3DShape dataset, along with the introduction of the reduced chi-square statistic for diversity measurements. The methodology design is substantiated by limited empirical results.

**Strengths:**

1.$\ $Some insights of this paper do exist in the generative model field. For instance, the widely adopted metrics FID and IS are indeed not tailored for diversity measurement, and the notion of "mode collapse" in many papers does lack a statistical definition. However, note the claims may still lack deeper consideration that these metrics are still meaningful and some thoughts have already existed in previous literature.

2.$\ $The proposed 3DShapesHD dataset and the reduced chi-square statistic contribute to the community.

**Weaknesses:**

1.$\ $There are several relevant works that are not being mentioned or fully discussed, which maintain high similarity to the proposed method.  [1] has already provided a more precise definition and analysis of mode collapse and proposes the mode completeness condition that requires consistent probability measures regarding samples. Moreover, in methodology, matching the real data's inverses with the latent prior is somehow a general design as similar mechanisms are adopted in [1,2,3,4]. There are also recent (in two years) efforts on solving mode collapse that need to be discussed and compared, including [1,5,6].

2.$\ $The claims about the existing metrics like "we found that these metrics correlate poorly with the reduced chi-square, which shows that they are not suitable for exact measurements" in Sec. 5.1 are confusing. Previous metrics like FID evaluate the distance between the feature distributions of the generated data and real data, which surely incorporate diversity evaluation as well as image quality and are meaningful. It is meaningless to evaluate diversity without a certain quality guarantee.

3.$\ $The empirical results are limited and confusing. 1) The comparison merely involves StyleGAN2 variants, leaving alone other recent efforts on solving mode collapse. 2) The results of precision and recall failed. It would be better to consider other metrics like density and coverage [7]. 3) FID results in Table 1 indicate that the proposed method is worse in generation quality even with an advantage in diversity, which makes the results somehow meaningless.

[1] IID-GAN: an IID Sampling Perspective for Regularizing Mode Collapse. IJCAI 2023.

[2] Vee-gan: Reducing mode collapse in gans using implicit variational learning. NIPS 2017.

[3] Mggan: Solving mode collapse using manifold-guided training. ICCVW 2021.

[4] Dist-gan: An improved gan using distance constraints. ECCV 2018.

[5] Improving Generative Adversarial Networks via Adversarial Learning in Latent Space. NeurIPS 2022.

[6] UniGAN: Reducing Mode Collapse in GANs using a Uniform Generator. NeurIPS 2022.

[7] Reliable fidelity and diversity metrics for generative model. ICML 2020.

**Questions:**

1.$\ $What is the fundamental difference between the proposed method and previous methods that match the real data's inverses with the latent prior?

2.$\ $What is new in this paper's analysis of mode collapse compared to that in [1].

3.$\ $With the sample numbers in the bins as a discrete distribution, several statistical metrics can be calculated. Why do you prefer the reduced chi-square statistic and how about the performance on other metrics like KL divergence?

4.$\ $Sec. 4.2 "If we are given the optimal weights of the convolution layers in E in advance, ...": It is a strong assumption. It appears hard to reach such optimality.

Please also see the weaknesses.

[1] IID-GAN: an IID Sampling Perspective for Regularizing Mode Collapse. IJCAI 2023.

**Details Of Ethics Concerns:**

No ethics concerns.

---

### Official Review · Reviewer_p6Dj · 2023-10-25

**Soundness:** 2 fair
**Presentation:** 1 poor
**Contribution:** 2 fair
**Rating:** 3
**Confidence:** 4

**Summary:**

This paper proposes a method for training an Adversarial Autoencoder and combining it with GANs. For the evaluation of the proposed model, this paper suggests the 3DShapesHD dataset.

**Strengths:**

-	This paper suggests a high-resolution version of 3DShapes dataset.
-	The related works are well organized.

**Weaknesses:**

-	The presentation and clarity of the paper could be improved.
-	The statements without support should be substantiated with appropriate references or experimental results.
-	The suggested reduced chi-square metric does not correlate with widely adopted evaluation metrics, such as FID, precision, and recall (Table 1). The proposed method achieves better results only in the reduced chi-square metric. This paper should provide a justification for the superiority of the reduced chi-square metric to claim the superiority of the proposed model.

**Questions:**

-	Please see the Weakness Section above.

---

### Official Review · Reviewer_e5hk · 2023-11-01

**Soundness:** 2 fair
**Presentation:** 2 fair
**Contribution:** 2 fair
**Rating:** 3
**Confidence:** 4

**Summary:**

This paper claims that although previous generative models could generate realistic images, they cannot guarantee that the generated data distribution matches the real data distribution. To address this challenge, this paper proposes an autoencoder-based adversarial training framework. The method first train an Adversarial AutoEncoder (AAE). Then, the AAE is fixed and used to produce generated data to train a GAN. The authors did experiments on the 3DShapes dataset and showed the $\mathcal{X}_{\nu}^2$ score is better than other GAN methods.

**Strengths:**

The authors proposed the AutoEncoder-coupled GAN (AE-GAN). Specifically, an AAE is first trained and fixed. The AAE is used to produced generated data for the AE-GAN. The AE-GAN has a new generator $G'$ that is to resemble the generator $G$ in AAE. The discriminator in AE-GAN takes two inputs, the real image and the outputted image from AAE, or the generated images by $G'$ and $G$. The AE-GAN seems novel to me.

**Weaknesses:**

- The claim is not well justified. The authors mentioned in the abstract that their proposed method ensures that the density of the encoder’s aggregate output distribution closely matches the prior latent distribution. However, there is neither theoretical justification nor sufficient empirical verification for this. I understand that the authors plotted in Fig. 2 of the distribution of bins, but still the curve produced by the proposed method does not match the theoretical curve. The experiment is only done on the 3DShapes dataset, and the latents have a uniform distribution. Experiments are not sufficient. The authors need to provide quantitative results to measure the overall performance of the proposed method. The method should be compared to StyleGAN in terms of generated images using FID and Inception Scores on large datasets such as LSUN, FFHQ and ImageNet datasets.

- Even for the 3DShapes dataset, the experimental results are not good. In table 1, while the method is slightly better than StyleGAN-XL in $\mathcal{X}_{\nu}^2$, but it is much worse than StyleGAN-XL in FID. Ultimately, we hope the generated images to be as realistic as possible.

- The organization of this paper is not good. The authors introduced the dataset and evaluation method in Sec. 3. I think the proposed method is specifically designed to address the challenges for the 3DShapes dataset. However, the 3DShapes dataset can be created automatically. A generative model is not necessary for generating these images. Why do we need to spend effort design specific methods for this task?

- Many parts of this paper require clarification:
    - Why "being able to generate every image in the data distribution does not imply reproducing the correct distribution"?
    - What data are used to produce Fig. 1? Are they two-dimensional toy data? Is the visualization in Fig. 1 in the data space or latent space? Looks like the discriminator is a linear classifier. Since the data points forms a circle distribution, I don't think a linear discriminator has sufficient capacity of differentiate between real data and generated data points. This experiment can not prove that oscillation problem is caused by adversarial training, but could be caused by discriminator's capacity.
    - "a recognition network that computes the six generating factors from an image". How accurate is that network? I think the performance of this network could impact the evaluation of different methods.
    - The authors mentioned: "Note that $x − y$ will always have a mean of 0 and thus cannot oscillate". What does oscillation mean for random variables? Why having a mean of 0 prevents oscillation?
    - In the experiments, the authors need to demonstrate how large batches improves the performance.
    - The well-structured latent space of $G$ is approximately a Gaussian distribution, since in Eq. 4 the authors regard the Gaussian distribution as real data. The well-trained encoder $E$ will encode the real data to a Gaussian distribution in the latent space. Why the authors call it well-structured?
    - While the discriminator in Eq. 7 accepts a pair of images is interesting, the authors need to justify the effectiveness of concatenating a pair of images along the channel dimension as input for the discriminator in Eq. 7.

**Questions:**

The authors need to address the concerns and questions in the Weakness part above.

---

### Official Review · Reviewer_Jjz9 · 2023-11-06

**Soundness:** 2 fair
**Presentation:** 2 fair
**Contribution:** 1 poor
**Rating:** 3
**Confidence:** 4

**Summary:**

The paper proposes a training framework aided by multiple autoencoder architectures that aim to align the aggregate encoded distribution with the prior latent law. The motivation stems from covering all modes of the input distribution by learning an accurate representation. Further, the reconstructed image quality is improved by coupling a GAN generator. The efficacy of the proposed method is tested using the reduced Chi-square goodness-of-fit test, based on generated observations from the 3DShapesHD dataset.

**Strengths:**

The organization of the article is good with ample discussion on related literature.

**Weaknesses:**

The paper's motivation seems unclear and divided between solving multiple problems that do not necessarily fall in line. The solution proposed also seems incremental with inadequate quantitative and qualitative evidence to show in favour. Also, there are numerous typographical/grammatical errors present in the article.

**Questions:**

1. It is perhaps not fair to say that mode collapse is not clearly defined in the literature. The article, later in the section 'Mode Collapse' mentions several prior works which prescribe remedies based on clear definitions.
In any case, the paper does not offer a solution.
2. What do the authors mean by 'uncommon features' [Abstract]? If it is uncommon in the sense that it carries little or no information, should we really need to bother? This seems crucial as the goal of representation learning--- which the encoders in autoencoders carry out during dimensionality reduction--- is to get rid of such features.
3. What do the authors mean by 'inversion task' [Introduction]? Also, can you explain what is meant by the `unlimited capacity' of neural networks?
4. [Section 4] Is there any theoretical/empirical guarantee to the claim that vanilla AEs are 'inherently resistant' to mode collapse? This question comes naturally based on the earlier statement "*being able to generate every image in the data distribution does not imply reproducing the correct distribution, which additionally requires that each image occur at the same frequency in the generated images as in the training data*". If the support of the data is non-convex, it is more likely to result in discontinuous decoder maps which in turn increases the chance of mode collapse (since neural networks fail to approximate discontinuous functions well [1]). Can the authors clarify?

[1] A Geometric Understanding of Deep Learning, Na Lei, Dongsheng An, Yang Guo, Kehua Su, Shixia Liu, Zhongxuan Luo, Shing-Tung Yau, Xianfeng Gu, in Engineering, 2020.

5. "*For example, consider training on a set of portrait photos...gains more hair, etc.*"Do you have any empirical or theoretical evidence to show for (e.g. experimental results or prior work)? Is this quite common in general?
6. What is meant by "*...the distribution of x is determined up to translation.*" and "*...and thus cannot oscillate*" [Section 4.1]
7. "*By adding a few more layers in the fully-connected part of E ... computationally cheap*" [Section 4.2]
Can the authors specify, perhaps with supporting experiments, the number of layers to be added to reach a required level of approximation capacity?
8. It seems difficult to know specifications regarding hyperparameter tuning, and exact model architectures readily from the paper, given a data set. A supplementary note on the same might be helpful.
Also, if the goal is to match the encoded distribution well with their latent counterparts, why not use Wasserstein Autoencodrs [2] instead of AAEs?

[2] Wasserstein Auto-Encoders, Ilya Tolstikhin, Olivier Bousquet, Sylvain Gelly and Bernhard Schoelkopf, International Conference on Learning Representations, 2018.

"*Theoretically, this simple procedure should suffice to ensure that the aggregate posterior distribution of the latent code is identical to the prior.*" Such a claim is never substantiated theoretically.